# Inflammasomes in the Pathophysiology of Aortic Disease

**DOI:** 10.3390/cells10092433

**Published:** 2021-09-15

**Authors:** Markus Wortmann, Andreas S. Peters, Philipp Erhart, Daniel Körfer, Dittmar Böckler, Susanne Dihlmann

**Affiliations:** Department of Vascular and Endovascular Surgery, University Hospital Heidelberg, Im Neuenheimer Feld 420, 69120 Heidelberg, Germany; markus.wortmann@med.uni-heidelberg.de (M.W.); Andreas.Peters1@med.uni-heidelberg.de (A.S.P.); philipp.erhart@med.uni-heidelberg.de (P.E.); Daniel.Koerfer@med.uni-heidelberg.de (D.K.); dittmar.boeckler@med.uni-heidelberg.de (D.B.)

**Keywords:** aorta, aneurysm, dissection, aortitis, PAU, IMH, inflammasome, AIM2, NLRP3

## Abstract

Aortic diseases comprise aneurysms, dissections, and several other pathologies. In general, aging is associated with a slow but progressive dilation of the aorta, along with increased stiffness and pulse pressure. The progression of aortic disease is characterized by subclinical development or acute presentation. Recent evidence suggests that inflammation participates causally in different clinical manifestations of aortic diseases. As of yet, diagnostic imaging and surveillance is mainly based on ultrasonography, computed tomography (CT), and magnetic resonance imaging (MRI). Little medical therapy is available so far to prevent or treat the majority of aortic diseases. Endovascular therapy by the introduction of covered stentgrafts provides the main treatment option, although open surgery and implantation of synthetic grafts remain necessary in many situations. Because of the risks associated with surgery, there is a need for identification of pharmaceutical targets interfering with the pathophysiology of aortic remodeling. The participation of innate immunity and inflammasome activation in different cell types is common in aortic diseases. This review will thus focus on inflammasome activities in vascular cells of different chronic and acute aortic diseases and discuss their role in development and progression. We will also identify research gaps and suggest promising therapeutic targets, which may be used for future medical interventions.

## 1. Introduction

Thoracic and abdominal aortic diseases pose a life-threatening condition affecting various congenital and acquired pathologies, either acutely or chronically. The global death rate from aortic diseases ranges from 2.5 to 2.78 per 100,000 inhabitants [1,2], with incidences of 5–10 per 100,000 person-years [3,4]. The contribution of inflammation to the different aortic pathologies has long been ignored or considered a bystander effect. Recent evidence from animal models, histological analysis of human aortic tissues, and cell culture experiments now propose a causative input of inflammatory mechanisms in both aortic aneurysms [5] and acute aortic syndromes [6]. Understanding these processes that ultimately lead to the destruction of the aortic wall provides the basis for future anti-inflammatory therapies that might complement or even replace current interventions, which are as yet restricted to surgical aortic repair by implantation of covered stentgrafts or synthetic grafts. Many recent reviews have summarized the role of inflammatory cells, cytokines, and autoimmunity within the aortic wall, predominantly focusing on atherosclerosis and aortic aneurysms [7,8,9]. This review aims to summarize and evaluate the role of inflammasomes in different aortic pathologies with respect to the aortic wall composition, aortic cell types, lessons from animal models, and their clinical impact. 

## 2. Inflammasomes—The First Level Response to Infection and Tissue Injury

Recent evidence has changed the view on inflammation from being considered a disease to an essential physiological mechanism in tissue repair [10]. Inflammation is an adaptive response of cells and tissues to different stress conditions and triggers, such as infections and tissue injury [11]. It comprises two phases, innate and adaptive response, which are mediated by a coordinated process of intracellular and intercellular events. These events form complex regulatory networks that may result in acute or chronic conditions [10,11]. Depending on the type of inducer, different molecular sensors and mediators are activated, resulting in different inflammatory pathways. Although initially detected in leukocytes, the main receptors of the innate immune system, such as toll-like receptors and receptors of inflammasomes, can be found in virtually any cell type in different compositions, including cells of the cardiovascular system [10,12]. 

Inducers may be pathogen associated molecular patterns (PAMPs), such as biomolecules derived from viruses and bacteria, or damage associated molecular patterns (DAMPs), such as particles and biomolecules released from damaged cells and tissues. Following stimulation by an inducer, inflammasome proteins aggregate to form (cytosolic) high-molecular weight signaling platforms, which are called inflammasomes [12]. 

Multiple distinct inflammasomes have been identified; the assembly of each is dictated by a unique pattern recognition receptor (PRR). To date, more than twenty different PRRs have been described, a subset of which have been confirmed to initiate inflammasome assembly. These include nucleotide-binding oligomerization domain, leucine-rich repeat-containing (NLR) family members NLRP1, NLRP3, NLRC4, NLRP6, NLRP7, and NLRP9b, as well as the DNA sensor proteins Absent in Melanoma 2 (AIM2), Pyrin, and IFI16 [12,13,14]. For assembly, the PRRs oligomerize and recruit multiple copies of the adaptor protein Apoptosis-associated speck-like protein with a caspase-1 recruitment domain (ASC), which on its part, bridges further association with inflammatory caspases. The full activation of the inflammasome complex finally results in the activation of Pro-Caspase-1 to its active form, which in turn enzymatically activates the Pro-interleukins (IL) -1β, -18 and several other cytokines of the IL-1 family. The mature interleukins may then be actively secreted from the affected cell through membrane pores via Gasdermin family members and result in pyroptosis, a programmed necrotic cell death [15]. In addition to the canonical inflammasome signaling, non-canonical inflammasomes involving other caspases, such as caspase-4, -5, -11, as well as caspase-8 have been described in some tissues [16,17]. The following chapters will summarize the current knowledge on inflammasome activities particularly in vascular (aortic) cells, infiltrating and peripheral leukocytes, and their contribution to different aortic diseases.

## 3. Composition and Function of the Aorta: Basics for Understanding Aortic Diseases

As mentioned in the previous section, inflammasome signaling occurs within the cytoplasm of cells, thereby starting an initially asymptomatic process that may spread across neighboring cells and tissues and finally become symptomatic by affecting the function of the complete aorta. Yet, how can inflammasome signaling trigger different diseases within the same tissue? Although there are still many knowledge gaps that limit the answer to this question, the architecture of the adult aorta may provide some indications and will be briefly introduced here.

The aorta follows the common structure of large elastic arteries with three main layers [18] (Figure 1). As the innermost layer, the tunica intima extends from the lumen to the internal elastic lamina, containing predominantly endothelial cells (EC), arranged in a single-layer. These cover the luminal surface in direct blood contact and regulate vasomotor tone, hemostatic balance, permeability, immunity, and cell proliferation. The internal elastic lamina consists of elastic fibers and separates the tunica intima from the tunica media. The media consists of a multi-layer of vascular smooth muscle cells (VSMC), alternating with elastin and collagen fibers. The VSMCs guarantee both contractility and secretion of extracellular matrix components. The principal task of the tunica media is to balance contraction and dilation in order to regulate blood flow. For the aorta in particular, the main function is to overcome the gap of central and peripheral vessel diameters. As another interlayer the external elastic lamina separates the media from the adventitia. The tunica adventitia consists of collagen and fibroblasts as well as of vasa vasorum and nerves or nutrition and regulation. 

Together with VSMCs, the elastic and collagen bundles form lamellar units and guarantee elasticity to regulate blood flow. The number of lamellar units and elastic capacity decreases from the aortic arch to the distal aorta. In addition, aging of the aorta leads to a higher collagen-to-elastin ratio with increased stiffness and pulse pressure [19]. Another important function of the aorta is the reception and regulation of systemic blood pressure via baroreceptors feedback loops (mainly in the ascending aorta and the aortic arch). High aortic pressure decreases the heart rate, as well as the systemic blood pressure, whereas low aortic pressure leads to opposite effects [18].

The natural process of aging results in diminished aortic resilience and tensile strength [20]. Aortic dilation and wall thinning with increased risk of dissection and rupture arise as a consequence. Atherosclerosis plays a decisive role in aortic disease formation, especially for chronic diseases. The clinical picture of aortic diseases is manifold and presents as congenital or acquired acute and chronic pathologies (Figure 2). The term covers thoracic and abdominal aortic aneurysms (TAA and AAA), aortic dissections (AD), intramural hematomas (IMH), penetrating atherosclerotic ulcers (PAU), traumatic aortic injuries (TAI), pseudoaneurysms, aortic occlusive, and thromboembolic as well as diseases with clinically apparent, systemic inflammations. AD, IMH, and PAU are subsumed under acute aortic syndromes (AAS) [3].

## 4. Inflammasomes in Abdominal and Thoracic Aortic Aneurysms 

Clinically, the term aneurysm describes a dilation of the vessel wall including all three layers. In general, it is defined by an 1,5-fold increase of the vessel’s original diameter [21]. Symptomatic AAA presents as abdominal or back pain or peripheral embolization and requires immediate treatment. Based on anatomical criteria and comorbidities, treatment comprises conservative, open surgical, or endovascular strategies [22].

Etiologically, AAA has long been considered a special form of atherosclerosis. Today, it is regarded as a chronic inflammatory disease, because the most prominent pathologic feature of human AAA is infiltration with inflammatory and immune cells, such as macrophages, neutrophils, mast cells, and B- and T-lymphocytes, which are mostly located in the outer aortic wall. These cells produce enzymes, cytokines and chemokines interacting with VSMC, EC and the extracellular matrix by autocrine, paracrine and juxtacrine pathways (reviewed in [23,24]). In addition, each of the cellular interactions induces complex cellular signaling pathways (recently reviewed in [25]). Consequently, the inflammatory response results in remodeling and progressive degeneration of the aortic wall, which may finally end in rupture. 

### 4.1. Inflammasome Genes and Proteins Are Expressed in Different Cell Types of Aortic Aneurysms

Inflammasome signaling in AAA is reported by an increasing number of studies (Table 1). After the discovery that NLRP3 inflammasomes are activated by cholesterol crystals and required for atherogenesis [26,27,28] it was assumed that the NLRP3 inflammasome might likewise participate causally in AAA. 

To elucidate the role of different inflammasome proteins in human AAA, we and others initially investigated their expression and intramural localization in the wall of AAA specimen that were received during open AAA repair. Immunohistochemical ana-lysis indicated a strong expression of ASC, Caspase-1, Caspase-5, NLRP3, and AIM2, predominantly in sporadic infiltrating lymphoid cells and in lymphoid aggregates located in the outer media and adventitia of AAA samples (Figure 3 and [29,30]). In line with these findings, the gene expressions of *ASC, CASP1, IL1B, NLRP3,* and *IFI16* were found to be increased in the aortic wall of AAA specimen, compared to controls [29,31,32]. 

A more detailed immunohistochemical analysis revealed significantly higher expression frequencies of NLRP3 and AIM2 proteins in leucocytes located in intimal atherosclerotic plaques of AAA samples than in plaques of healthy aortas [30]. Moreover, within the AAA specimen, the inflammasome proteins NLRP3, AIM2, ASC, Caspase-1, and Caspase-5 were less frequently found in late stage lymphocytic aggregates within the media and adventitia than in intimal atherosclerotic plaques of early stage AAA samples [30]. The findings suggest that inflammasome activities in infiltrating leukocytes occur early during AAA development and decrease with AAA progression. This is in line with the hypothesis that AAA may be considered an autoinflammatory or autoimmune disease [33].

The activation of inflammasome signaling was also detected in peripheral blood mononuclear cells (PBMC) of vascular patients [34,35,36,37]. *CASP1* and *IL1B* gene expressions are increased in PBMC derived from male AAA patients, compared to the matched PBMC of male non-AAA patients. In addition, AIM2 and active Caspase-1 (p10), but not NLRP3 protein levels are significantly higher in PBMC of AAA patients along with increased Il-1α and Il-1β plasma levels [35]. Immunophenotyping by multicolor flow cytometry revealed that particularly AIM2 but not NLRP3 protein expression is significantly increased in peripheral granulocytes, monocytes, B- and T- lymphocytes of AAA patients. Accordingly, PBMC from AAA patients released significantly more IL-1β into the supernatant than PBMC from control patients, upon in vitro stimulation with foreign DNA, whereas LPS-induced NLRP3 inflammasome signaling did not show any difference in IL-1β release [34]. In summary, this points to a particular role of DNA-induced inflammasome activity via AIM2 in PBMC of AAA patients. 

Interestingly, the gene expression of *AIM2, NLRP3, CASP1, CASP5* and *IL1B* in PBMC generally rises with age in vascular patients [36], indicating a relationship between increased inflammasome signaling, aging, and aortic diseases. 

The expression of inflammasome components in thoracic aortic aneurysm (TAA) is less well documented. Similar to AAA, TAA is characterized by progressive dilation of the aortic wall. The molecular mechanisms underlying TAA however, differ from those in AAA. More than 30 hereditable disorders are associated with thoracic aneurysm, resulting—if not repaired–in dissection [38]. Accordingly, most studies on molecular mechanisms do not differentiate between TAA and thoracic aneurysm and dissection (TAAD). The pathological molecular pathways in familial TAA/TAAD are predominantly caused by mutations affecting genes that encode extracellular matrix proteins and members of the TGF-β signaling pathway (reviewed in [38,39]).

Inflammatory cell infiltrates and an increased production of inflammatory cytokines have been described in acute and familial TAA and in patients with Marfan syndrome [40,41]. In addition, Il-1β protein levels are dramatically increased in human TAA/TAAD, compared to healthy thoracic aortas [42,43]. Within the aortic wall of sporadic TAA/TAAD, an increased degradation of the contractile proteins myosin heavy-chain, and tropomyosin was observed in VSMC. Concurrently, the protein levels of NLRP3, ASC, Procaspase-1 and active Caspase-1 (p10 and p20) were increased in diseased thoracic aortic sections, particularly in macrophages and VSMC [44]. Thus, the NLRP3 inflammasome cascade appears to be as well associated with sporadic TAA/TAAD.

### 4.2. Pre-Clinical and Functional Studies Suggest a Causal Role of Inflammasome Signaling in the Development of Aortic Aneurysm 

The current knowledge on the pathogenic role of inflammasome signaling in development and progression of AAA and TAA/TAAD comes from different rodent models and from in vitro cell culture experiments using inflammatory cells, endothelial cells (EC), and VSMC (Table 1).

In addition to their expression in inflammatory cells, inflammasomes are inducible in vitro by different stimuli in vascular cells. Many years before inflammasomes were detected, Libby et al. described induction of IL-1 gene expression by bacterial endotoxins in human EC and VSMC [45]. Since then, a lot of other stimuli have been shown to induce different inflammasome components in these cells. The administration of the calcification inducer β-Glycerophosphate to primary VSMC can induce *NLRP3 (NALP3), ASC* and *CASP1* gene expressions, resulting in increased IL-1β secretion [46]. Moreover, *AIM2* can be induced by tumor necrosis factor (TNF)-α, Interferon (IFN)-γ and by synthetic DNA (poly-dA:dT) in human aortic EC and VSMC, and AIM2 protein expression is detectable in VSMC of human aortic lesions [29]. In vitro cultured VSMC derived from human AAA were shown to express ASC and to respond with induction of AIM2, NLRP3, IFI16, and Caspase-1 proteins upon exposure to necrotic cell debris from neighbouring cells, which was further increased by simultaneous stimulation with IFN-γ [47]. The treatment of human AAA-derived VSMC with Angiotensin II (AngII) results in induction of ASC and the DNA sensor IFI16, whereas overexpression of IFI16 in AngII-treated rat VSMC enhances ROS levels, activates Caspase-1, promotes migration and triggers release of IL-1β [31]. In line with these reports, *Nlrp3* gene deletion in mice attenuates AngII- induced Nlrp3 inflammasome activation, phenotypic transformation from a contractile to a synthetic phenotype and proliferation of primary aortic VSMC [48]. Together, these findings clearly indicate that inflammasome signaling in EC and VSMC may contribute to sterile inflammation in AAA. 

Studies with different animal models suggest that inflammasome signaling is causally involved in the formation of aortic aneurysms. The application of periadventitial elastase to thoracic aorta in wild-type mice results in progressive dilatation of the aorta with elastin fragmentation, VSMC loss, macrophage infiltration, and increased Il-1β expression, thus presenting features consistent with human TAA [43]. The genetic deletion of Il-1β and Il-1 receptor in this model significantly decreased thoracic aortic dilation, preserved elastin and VSMC and reduced infiltration with inflammatory cells. In addition, when wild-type mice were treated pharmacologically with Il-1β inhibitors, TAA formation and aortic dilation were reduced [43]. Similar results were reported from another established mouse model of aortic aneurysm and dissection, which induces dilation of the thoracic and abdominal aorta by continuous infusion with AngII in *Apoe*^−/−^ mice for 28 days. Again, inhibition of Il-1β by Il-1 receptor antagonist (Il-1Ra) or an anti-Il-1β antibody significantly suppressed aneurysm formation after AngII infusion [49], indicating that Il-1β might be a promising target for TAA and AAA treatment.

Using the same mouse model, Usui et al. demonstrated that the Nlrp3 inflammasome in adventitial macrophages is causally involved in AngII-induced AAA formation [50]. Genetic deficiency in *Nlrp3*, *Asc,* or *Casp1* in *Apoe*^−/−^ mice significantly decreased the incidence, maximal diameter and severity of AAA along with reduced adventitial fibrosis, inflammatory cell infiltration and cytokine expression. Functional in vitro studies revealed that activation of the Nlrp3 inflammasome is caused by mitochondrial oxidative stress in adventitial macrophages in this model [50]. In line with these findings, Nlrp3 inflammasome activation of adventitial macrophages was shown to mediate hyperhomocysteinemia-aggravated AAA in two different mouse models (AngII-induced AA in *Apoe*^−/−^ mice, and calcium phosphate induced AA) [51]. Again, increased mitochondrial oxidative stress in macrophages was observed in hyperhomocysteinemia-aggravated aneurysms, and blocking of this stress abolished inflammasome activation [51]. 

Finally, the genetic deletion of IL-18 in AngII infused mice (WT background) attenuated the devlopment of AA by reducing macrophage infiltration and by altering macrophage polarization in abdominal aortic tissues [52]. In additon to using AngII for induction of aneurysms, all animals were additionally treated with beta-aminopropionitrile (BAPN) in this study.

Besides in macrophages, inflammasome activation in VSMC appears to interfere with aortic aneurysm formation in mice. We recently demonstrated that *Aim2* deficiency affects the phenotype of murine aortic VSMC, thereby reducing AngII-induced formation of aortic aneurysm in six month old mice [53]. Compared with VSMC from wild-type mice, VSMC isolated from *Aim2^−/−^* mice were larger, less viable, and underwent stronger calcification in mineralization medium, along with reduced inflammasome gene expression. Similar to the observation in macrophages, oxidative stress in mitochondria of VSMC is associated with the induction of inflammasome proteins in AAA tissue. By using two wild-type strains, differing in the function of the mitochondrial gene nicotinamide-nucleotide-transhydrogenase (Nnt), we found increased protein expression of Nlrp3, Aim2, Asc and Caspase-1 in VSMC of the Nnt-deficient mouse strain, along with higher levels of oxidative DNA damage and an increased incidence of AngII-induced aortic aneurysm [54].

Mechanistically, Nlrp3-caspase-1 inflammasome signaling in VSMC was also associated with degradation of contractile proteins in a mouse model, where TAA/TAAD was induced by a combination of AngII-infusion and high fat diet [44]. Challenged wild-type mice showed increased degradation of myosin heavy chain and tropomyosin, whereas this was not observed in challenged *Nlrp3*- and *Casp1*-deficient mice. The data suggest that the activation of the Nlrp3-Capsase-1 inflammasome cascade promotes the degradation of contractile proteins in VSMC, thereby leading to TAAD development.

Taken together, there is strong evidence that inflammasome activation is induced by increased oxidative stress in both macrophages and VSMC, thereby promoting aortic aneurysm formation (Figure 4).

### 4.3. Targeting Inflammasome Signaling for Non-Invasive Treatment of Aortic Aneurysm and Dissection

Recent progress in understanding the mechanisms of inflammasome signaling in VSMC and inflammatory cells has enabled the identification of putative targets to retard or even prevent aortic remodeling, dilation, and/or rupture by non-invasive therapies.

MCC950 is a potent and specific inhibitor of the NLRP3 inflammasome, blocking the release of IL-1β induced by NLRP3 activators, such as nigericin, ATP, and MSU crystals [55]. In both male and female mice challenged with high fat diet and AngII infusion, MCC950 was recently shown to prevent aortic destruction and aneurysm formation in different aortic segments [56]. Consistent with this effect, MCC950 preserved elastic fibers and reduced apoptosis and protein degradation in VSMC. In addition, the challenge-induced Mmp-9 activity in macrophages was shown to be directly caused by Caspase-1 activity, which was significantly reduced by MCC950 administration [56]. Although the direct contribution of the NLRP3-Caspase-1 inflammasome to MMP-9 activation remains to be demonstrated in human aortic aneurysms, targeting this pathway with MCC950 is a promising approach for preventing aortic destruction.

A different approach to test inflammasome inhibition for treatment of aortic aneurysms and TAAD was recently described by Le et al. [57]. The authors hypothesized that pyruvate kinase M2 (PKM2)-dependent glycolysis regulates IL-1β secretion via the activation of inflammasomes in TAAD. PKM2 is a rate limiting enzyme in glycolysis, the level of which was significantly elevated in aortic tissues from TAAD patients and from a previously established beta-aminopropionitrile (BAPN)-induced mouse model for TAAD [57,58]. Treatment of BAPN-challenged mice with TEPP-46, a potent activator of PKM2, resulted in the attenuation of TAAD, decreased aortic diameter, lowered macrophage infiltration, and reduced NLRP3 inflammasome-mediated IL-1β secretion [57]. Accordingly, targeting PKM2 and inflammasome activity by TEPP-46 might be an additional strategy to delay TAAD progression.

## 5. Inflammasome Activity in Acute Aortic Syndromes (AAS): Thoracic Dissections, Intramural Hematoma, Penetrating Aortic Ulcers, and Traumatic Aortic Injury

Acute aortic syndrome (AAS) describes a clinical diagnosis with acute pain due to a loss of the aortic wall’s integrity. It comprises aortic dissection (AD), intramural hematoma (IMH), and penetrating aortic ulcers (PAU), which represent different types of wall lesions requiring diagnostics and medical care [59] (Figure 2).

### 5.1. The Role of Inflammasomes in Aortic Dissections (AD)

Aortic dissection is caused by a tear of the aortic intima, allowing blood to form a new path by separating the layers of the aortic wall and dividing the lumen of the aorta in a true and a false lumen. Without treatment, rupture or malperfusion of the aorta and emitting arteries may follow. It predominantly affects the descending thoracic aorta, whereas isolated abdominal manifestations of AD are rare.

In comparison to aortic aneurysm with an overall incidence of around 250 to 650 cases per 100.000 patient-years, aortic dissections occur less frequently (2.6 to 3.5 per 100.000 patient-years) [60,61]. Consequently, information about the role of inflammasomes in the pathophysiology of aortic dissection is sparse, compared to aneurysms. This holds true for acute as well as for chronic expanding aortic dissections. In addition, as mentioned above, most animal studies on molecular mechanisms do not differentiate between TAA and TAAD.

Similar to aortic aneurysms, the NLRP3 inflammasome is also the most widely investigated inflammasome in aortic dissection in human tissue samples (Table 1). The NLRP3 protein was found to be highly expressed in tissue samples of 10 patients with descending aortic dissections, mainly in dedifferentiated vascular smooth muscle cells within the media and in macrophages within the adventitia [62]. In vitro, inducing metabolic stress in aortic smooth muscle cells with palmitic acid upregulated the expression of NLRP3 and induced activation of metalloproteinase 9 (MMP9), whereas knocking down of NLRP3 decreased MMP9 cleavage and improved smooth muscle cell function [62].

In a small cohort of six patients with sporadic ascending aortic dissection, the expression of the AIM2 inflammasome protein was markedly increased in tissue samples of the aortic wall in comparison to controls with ascending aortic aneurysms or controls without any aortic disease [63]. AIM2 expression was mainly located in smooth muscle cells of the media and adventitia.

IL-1β was significantly higher expressed in the media of 10 patients with acute aortic dissection, both on mRNA as well as protein levels [42]. Interestingly, IL-1β expression was not only elevated compared to control patients, but also compared to patients with aortic aneurysm.

IL-18 expression was found to be significantly increased in plasma and in aortic samples from human AD patients [64]. Functional analysis revealed that aortic IL-18 was mainly derived from macrophages and also partly derived from CD4+ T lymphocytes and vascular SMCs. This suggests that IL-18 may participate in AD by regulating macrophage differentiation and macrophage-induced SMC apoptosis.

Functional studies on the role of inflammasomes in aortic dissections are hampered by the lack of a specific animal model that uniquely mimics aortic dissection. Although in humans, the etiopathology of aortic dissection is clearly different from aortic aneurysm, the most widely used rodent model for aortic dissection is the AngII mouse model that was already described above as a model for aortic aneurysms. There, aortic disease is induced by a continuous application of AngII—mostly via a subcutaneously implanted osmotic pump. To increase the incidence of aortic disease, several modifications are applied. For example, use of male mice [65], high fat diet (western diet) or genetic modifications such as low-density-lipoprotein (LDL) receptor or apolipoprotein E knockout [66] increase the incidence of aortic aneurysms, without providing explicit insight into the dissection process. Consequently, the applicability of the AngII model for investigating molecular mechanisms in the degenerating aorta is controversially discussed in the literature, because it depicts morphological features of both aortic aneurysm and dissection [67]. In line with this, most findings regarding inflammasome activity in aortic aneurysms and dissections in rodent models display great overlap and the disease is often termed TAA/TAAD.

As described in the previous chapter, the AngII mouse model was used to demonstrate that Nlrp3 signaling contributes to the degradation of contractile proteins in VSMC in TAA/TAAD. Both smooth muscle actin degradation and the incidence of aortic dissections was reduced in *Nlrp3* and *caspase-1* knockout mice, as well as by treatment with glyburide, which is an antidiabetic drug incidentally inhibiting Nlrp3–caspase-1 inflammasome complex formation [44].

Moreover, the inhibition of the Nlrp3 inflammasome by the specific inhibitor MCC950 significantly reduced aortic destruction, contractile protein degradation, and cell-death of VSMC in this mouse model, resulting in a reduction of TAA/TAAD incidence [56]. This effect was mainly attributed to a reduced cleavage of the N-terminal inhibitory domain of matrix metalloproteinase 9 by caspase-1, suggesting a direct activation of matrix metalloproteinase 9 by the Nlrp3 inflammasome.

Although it is not clear whether these mechanisms are causally involved in a similar way in human TAA or AD, there is gathering evidence that VSMC dysfunction and dysfunctional mechanosensing of the aortic wall are crucial features in the pathophysiology of both aortic aneurysms and dissections [68], and these findings emphasize the potential importance on NLRP3 in this context.

The same is true for the second TAA/TAAD mouse model described in the previous chapter, where the lysyl oxydase inhibitor β-aminopropionitrile monofumarate (BAPN) was administered for four weeks via drinking water for disease induction [58]. In wildtype C57Bl/6 mice BAPN treatment resulted in an incidence of aortic dissection of 87% with a relevant lethality from aortic rupture (45%). The aorta in this model is enlarged from the root to the thoracic segment, sometimes even to the abdominal segment with destruction of elastic fibers and aortic smooth muscle cells. Altogether, this fairly accurate resembles many important histopathological findings specific for aortic dissection in human patients, in contrast to the AngII-induced mouse model. Using the BAPN model, Le et al. examined the role of glycolytic enzyme pyruvate kinase M2 (PKM2) in aortic dissection [57]. PKM2 indirectly mediates Il-1β secretion via aerobic glycolysis and has shown to be protective in several diseases with a chronic inflammatory strain. Regarding aortic dissection in the BAPN model, activation of PKM2 suppresses Nlrp3 mediated Il-1β secretion and, in this way, both reduced lethality due to aortic rupture as well as the maximal aortic diameter.

### 5.2. Inflammasome Activities in Intramural Hematoma (IMH), Penetrating Aortic Ulcers (PAU) and Traumatic Aortic Injury (TAI)

To our knowledge and after profound literature search, the role of inflammasomes in other acute aortic syndromes (IMH, PAU and TAI) has not been investigated so far. Reasons for this lack of knowledge might be the low incidence of these diseases and the lack of appropriate animal models. Another reason might be the unavailability of tissues derived from these patients, because they are either treated conservative, or interventional, particularly by thoracic endovascular repair (TEVAR) [22]. Finally, the acute, often emergency situation of the patients does usually not allow collection of blood or tissue for research purposes.

IMH is a life-threatening acute aortic disease, representing 6–10% of all aortic syndromes [69]. Just like AD, IMH predominantly affects the thoracic aorta, whereas abdominal manifestations are rare. In contrast to AD, neither an intimal tear nor a false lumen develops in the aortic wall in the case of IMH. Instead, it is a contained hematoma resulting in bleeding within the medial layer that weakens the aortic wall. The leak is limited by the surrounding tissue pressure and appears as dilatation of the vessel. Early phases of IMH can regress or progress to AD or rupture. In contrast, long-lasting IMH can progress to TAA or pseudoaneurysm, a false aneurysm in which injury to all three layers of the wall results in blood leakage [70,71]. Because hypertension and the hypertensive factor AngII can induce IMH in combination with aneurysm in mice, the previously described AA/TAAD mouse model using osmotic pumps for subcutaneous infusion of AngII might be useful to investigate the role of inflammasomes in IMH development, as well [72].

PAU is defined by an ulceration of an aortic atherosclerotic plaque penetrating through the internal elastic lamina into the aortic media, causing 2–7% of all acute aortic syndromes [69]. PAU are often connected to atherosclerotic AAA in elderly patients. Its etiology is unclear, however, there is some evidence for vessel wall inflammation to be associated with disease progression [73]. Thus, given that animal models or tissues will be available, it might be interesting in the future to investigate participation of inflammasome signaling in aortic cells during PAU development and progression.

TAI is a blunt trauma, often as a result of rapid deceleration associated with traffic accidents or falling from great heights. In 90%, TAI is located at the aortic isthmus [74,75]. Accordingly, it may be hypothesized that inflammasome signaling might at best play a role during healing of the injured aortic tissue after intervention.

## 6. Inflammasomes in Aortic Atherosclerosis and Aortic Occlusive Disease (AOD)

Atherosclerotic lesions result from a lipid-driven, chronic inflammatory process of the arterial intima, that may progress to macrovascular pathologies, resulting in hypoperfusion and ischemia. Within the aortic wall, decrease of the aortic lumen by atherosclerotic plaques may cause organic damage (e.g., renal) or claudication symptoms. Moreover, thromboembolic events can lead to a distal perfusion deficit with life-threatening consequences, i.a. stroke, mesenteric or critical limb ischemia.

Aortic occlusive disease (AOD) is generally caused by atherosclerosis and is considered a subdiagnosis of peripheral arterial disease (PAD). Its central location within the aorta is associated with an early onset of disease and female sex, the reasons for the latter being rather unclear. This is in contrast to the more predominant involvement of the lower extremities in PAD where patients are mainly old men [76,77]. Much rarer causes for AOD are Buerger’s and Takayasu’s disease, both inflammatory vasculitides that are associated with an even earlier disease onset [78] (see below).

Already more than ten years ago it was described that cholesterol crystals—known before to show a late appearance in the necrotic core of mature atherosclerotic lesions—induce early inflammation by triggering the NLRP3-inflammasome. Hence, the activation of the NLRP3-inflammasome was regarded to take a key-role in atherogenesis [26,27]. Since then, the number of publications, reporting further details on inflammasome signaling in atherogenesis has multiplied and many review articles have summarized its pathological impact and possible therapeutic targets (i.e., [79,80]). Since the majority of these reports focused on cardiomyocytes and cardiovascular disease in general, we here limit our overview on studies investigating the role of inflammasomes on aortic atherosclerosis (Table 1).

*NLRP3* gene expression was found to be significantly increased in aortic full vessel-wall sections from patients suffering from either AOD or AAA compared to healthy controls. Interestingly, the gene expression of *NLRP1* was particularly increased in AOD. The authors concluded that activation of the NLRP3-inflammasome provides a general pattern in the inflammatory process of the underlying atherosclerosis, whereas the NLRP1-inflammasome seems to be specific for the pathogenesis of AOD [32]. The contribution of the NLRP3-inflammasome to the process of atherosclerosis and its putative role as a therapeutic target is widely accepted [80,81,82]. While the NLRP3-inflammasome is attributed to metabolic dysfunction, the NLRP1-inflammasome is usually linked to autoimmunity [83]. Bleda et al. provided data to support the idea that the NLRP1- and not the NLRP3-inflammasome leads to a proinflammatory shift of endothelial cells in PAD [84].

Obviously, the induction of the NLRP3-inflammasome does not only occur locally at the site of mature atherosclerotic lesions; rather, it is part of a systemic effect. This has been demonstrated by the overexpression of the NLRP3-inflammasome in macroscopically intact aortic tissue of patients suffering from coronary heart disease receiving aorto-coronary bypass surgery compared to healthy aortic tissue from kidney-donors [85]. In 2020 Chen et al. used an LDL-receptor-knock-out mouse-model to demonstrate that the Nlrp3-inflammasome plays a more prominent role in the atherosclerotic process in female mice. The difference was attributed to contrary effects of sex-hormones [86]. The latter could in part explain the association of increased incidence of AOD and female sex [76,77].

## 7. Inflammasomes in Vasculitis of the Aortic Wall: Aortitis, Takayasu Arteritis, Buerger Disease, Behcet’s Syndrome, Kawasaki Disease, and Reactive Arthritis (Reiter’s Syndrome)

The term aortitis describes a general inflammation of the aortic wall, which can be of both infectious and non-infectious origin. The latter includes not pathogen but autoimmune triggered vasculitis such as giant cell and Takayasu arteritis and less common Behçet’s, Buerger, and Kawasaki disease as well as Reactive Arthritis. Aortic dissection, rupture or thromboembolic events may develop as consequences of aortitis. Treatment is generally conservative with corticosteroids for non-infectious diseases and antibiotics for infectious diseases. Despite the obvious association of aortitis with PAMPs and DAMPs, only preliminary data is available on the role of inflammasomes during this process (Table 1).

With respect to Buerger’s disease the NLRP3-inflammasome has been proposed to be involved in disease progression. Data supporting this hypothesis are derived from a rat-model, where Buerger’s disease was mimicked by the injection of lauric acid into the femoral artery to cause segmental inflammatory thrombotic occlusion of lower extremity vessels [87,88]. No data from human patients are available so far, supporting a causative role of inflammasome signaling here.

Takayasu arteritis (TAK) is an autoimmune systemic arteritis of unknown pathogenesis. It particularly affects middle-aged women with an Asian background and presents itself with inflammatory lesions mainly localized in the aorta and its branching arms [89]. In a Japanese collective of nearly 100 patients with Takayasu arteritis, a single nucleotide polymorphism (SNP) in the MLX- gene was detected to be associated with severity and extend of the disease. This SNP results in a missense mutation causing a Q139R substitution in the transcription factor MAX (Max-like protein X). The authors could show that the mutation induced an upregulation of the NLRP3-inflammasome in VSMC of patients compared to healthy controls. Moreover, protein levels of NLRP3 and caspase-1 were significantly elevated in PBMC derived macrophages of TAK patients. The latter was paralleled by an increased level of the respective downstream mediator IL1β in the supernatant of cultured PBMC derived macrophages [89].

Behçet’s syndrome is a systemic inflammatory disease at the crossroad to autoimmune disorders that is possibly triggered by an aberrant response to infectious stimuli. Manifestations include many different sites of the body, among others vasculitis of arteries, such as the aorta [90]. Both mRNA and protein levels of NLRP3, ASC, and Caspase-1 were shown to be upregulated in PBMC and in skin lesions of patients with Behcet’s syndrome, compared to healthy controls. Upon in vitro stimulation of PBMC with LPS/ATP, the mRNA and protein levels of NLRP3, ASC and/or Caspase-1 were significantly up-regulated only in PBMC from active Behcet’s syndrome patients compared to healthy controls. This was accompanied by increased secretion of IL-1β from stimulated PBMC of the patients [91]. Although this clearly indicates a significant role of NLRP3 inflammasome signaling in Behcet’s syndrome, no data is available about inflammasome activity within the aortic lesions of these patients, so far.

Kawasaki disease (KD) is an acute systemic vasculitis in infants and children that particularly affects the coronary arteries. In addition, the aorta may be affected by inflammatory cell infiltration that involves all layers of the vessel wall [92]. Even after regression of the vasculitis, the biophysical properties of the aorta may be altered, resulting in aortic dilation or even AAA [93]. In a murine model of Lactobacillus casei cell-wall extract-induced KD, where aortitis was also associated with AAA development, a causative role of Il-1β and the Nlrp3 inflammasome was demonstrated [94,95]. In line with these findings, Il-1β production, caspase-1 activation, leukocyte infiltration, and fibrotic changes in the aortic root were detected in a second murine model of KD, induced by Candida albicans water-soluble fraction. Here, caspase-1 activation and Il-1β production were shown to occur in bone marrow-derived dendritic cells, which was significantly inhibited by a deficiency of Il-1β, Nlrp3, and Asc [96] or by an antibody targeting Il-1β [97].

Reactive arthritis (Reiter’s syndrome) develops in response to gastrointestinal or genitourinary infection with bacteria, such as Chlamydia, Salmonella, Shigella, Yersinia, or Campylobacter [98]. The disorder, which is usually characterized by arthritis, urethritis, and conjunctivitis, manifests within a few days to weeks after infection and is considered an aberrant autoimmune response. It preferably affects men aged 20–40 with a genetic predisposition in immunological factors [98]. Clinical manifestation within the aorta has been documented in a few case reports [99,100]. So far, no association with inflammasome signaling has been reported.

## 8. Alterations in Inflammasome Genes and Their Association with Aortic Disease

Several genetic disorders can influence the aortic wall structure or cause aortic abnormalities. Most of the affected genes encode proteins of the connective tissue or extracellular matrix components and predominantly result in diseases of the thoracic aorta [101]. Although genetic studies have provided evidence of the key role that inflammasome-related genes play in the pathogenesis of autoinflammatory diseases, very little is known about the association of genetic variations in inflammasome genes and aortic disease.

In 2011, a study by Roberts et al. assessed, whether there is genetic evidence for a role of NLRP3 in AAA [102]. By genotyping 1151 AAA patients and 727 controls for two inflammasome-related single nucleotide polymorphisms (SNPs), the authors found a significantly higher concentration of plasma IL-1β in participants who were homozygous for the common C allele of the *NLRP3* gene (*SNP rs3582941)*. The data suggested that genetic variability within the NLRP3 inflammasome may be important in the pathophysiology of AAA.

By using a next-generation sequencing approach, a mutational profile of rare variants in inflammasome-related genes was identified for Behçet disease [103]]. Among the pathogenic alleles identified, the major contributor was Nucleotide-Binding Oligomerization Domain Protein 2 (NOD2), one of the pattern recognition receptors, responsible for the activation of NLRP3 and others.

To our knowledge, no data is available so far on the association between other aortic diseases and genetic alterations in the inflammasome genes *IL1B, CASP1, CASP5, ASC, AIM2, NLRP1,* and *NLRC4,* or others.

## 9. Summary and Conclusions

Inflammasome signaling has been demonstrated in response to a number of external and internal stimuli in aortic cells. For both acute and chronic aortic diseases, different stimulating components have been identified, which act as PAMPs and DAMPs on aortic cells, thereby activating specific TLRs and inflammasome receptors. Following assembly to a multiprotein complex, containing ASC and Caspase-1 filaments, a common signaling pathway is activated within the affected cells, resulting in the release of inflammatory cytokines from the aortic or tissue infiltrating inflammatory cells. At first, the cytokines, particularly IL-1β and IL-18, act locally on neighboring cells and depict the first step to communicate a microlesion to the tissue environment.

Despite the clinically different appearance of aortic diseases, this suggests a common biological mechanism that may initiate a cascade of inflammatory processes following microlesions. These inflammatory processes may proceed subclinically, or progress to symptomatic events, depending on the capabilities of the affected cells and tissues to interfere with the immune system, and to repair the microlesions. It is tempting to speculate that most of the microlesions heal unnoticed, whereas only a subset of the initially silent inflammasome activations may provoke a chronic autoinflammatory process or transit to autoimmune responses. This ultimately leads to the question, which events decide about further progress to pathological conditions, resulting in the different clinical manifestations. The answer to this question is still unknown and completely speculative. The individual composition of additional risk factors, such as genetic predisposition, lifestyle, and accidents, as well as the site of origin of a certain microlesion might impact the final outcome.

Another open question refers to the particular sources of the aortic microlesions and their functional interplay with inflammasome signaling in the absence of pathogens. During aging, different molecular changes accumulate within all tissues, including the aorta. As described in the previous chapters, cholesterol-driven NLRP3 inflammasome activation is a causative factor on atherosclerosis and occlusive aortic disease [26]. Other age-related senescent pathways could also result in inflammasome-activating microlesions within aortic tissue, e.g., elastin breakdown, accumulation of advanced glycation endproducts/their receptors (AGE/RAGE), lysosome accumulation of lipid peroxidation adducts, etc. Reports from non-vascular studies indicate that elastase effects could be causative to inflammasome activation [104]. Elastin fragments could induce NFkB activation and IL-1β upregulation—a prime condition for inflammasome activation [105]. Moreover, mitochondrial derived oxidative stress that was found to induce an inflammsome response in macrophages [50] and VSMC [54] of mouse aneurysms, is expexted to increase during aging, thereby creating an adverse feed forward (vicious) cycle of inflammation and oxidative stress [106]. Similarly, AGEs react with their receptor RAGE to activate NF-κB-driven inflammation and oxidative stress in vascular cells, which further increases production of AGEs [107]. Taken together, several pathways and interactions could be mutually causative in self-perpetuating processes, in which inflammsome signaling is only one (important but not the only causative) player of age-related aortic disease.

Just as for other inflammatory disorders, the most extensively investigated inflammasome receptor throughout all aortic diseases, is NLRP3, although AIM2 and NLRP1 appear to play specific roles in AAA and/or AOD, respectively. The picture how these inflammasome receptors may trigger destruction or repair of the aortic wall is far from being understood. Existing data are limited on canonical inflammasome signaling in aortic cells, whereas the role of other inflammasome receptors, such as NLRC4, NLRP6, NLRP7, NLRP12, DDX58, IFI16 and others, has not been studied yet in aortic cells.

Irrespective of further details that need to be investigated to fully understand the mechanisms within aortic wall lesions, the common nature of inflammasome signaling as an initiating step of inflammation, healing, and remodeling holds promise for the development of new, non-invasive therapies, targeting NLRP3, AIM2 and/or IL-1β. The major challenge will be to identify compounds and dosages, which might reduce the overshooting inflammasome signaling in response to sterile inducers, while not interfering with healing or pathogen defense during infection. Some of the details summarized in this review might help to identify such therapeutics, for reducing aortic aneurysm growth, halting dissection, or preventing rupture.

## Figures and Tables

**Figure 1 cells-10-02433-f001:**
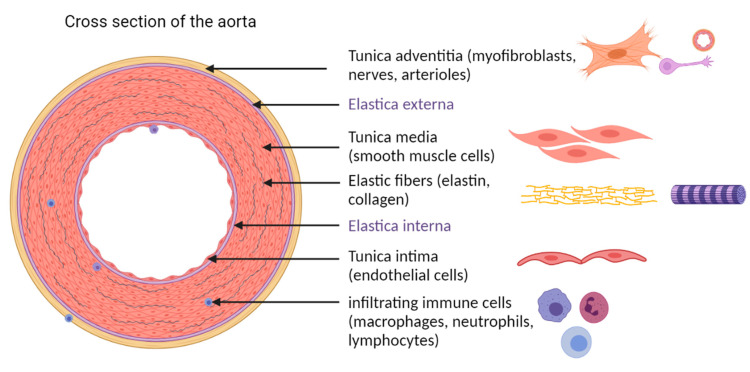
Cross section and cell composition of the aortic wall. As described in the text, the aortic wall is composed of three layers: the tunica intima, comprising a monolayer of EC, the tunica media, built of multiple VSMC layers, and the tunica adventitia, comprising myofibroblasts, nerves, and the vasa vasorum with arterioles. The Elastica externa and the Elastica interna, separating the three layers, are made of elastin and collagen bundles. Single infiltrating immune cells, such as macrophages, neutrophils and lymphocytes, spread within the complete aortic wall. The figure was created with BioRender.com (accessed on 14 September 2021).

**Figure 2 cells-10-02433-f002:**
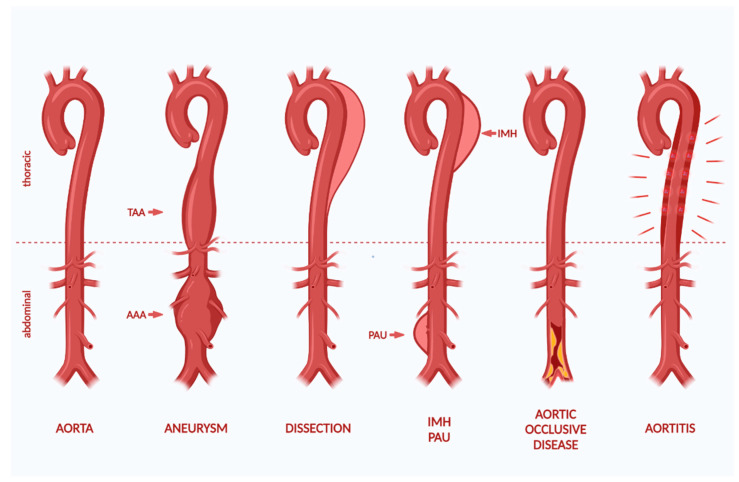
Schematic representation of a healthy aorta and aortic diseases. From left to right: healthy aorta, thoracic and abdominal aortic aneurysm, aortic dissection, intramural hematoma and penetrating aortic ulcer, aortic occlusive disease, and aortitis. The figure was created with BioRender.com (accessed on 15 September 2021).

**Figure 3 cells-10-02433-f003:**
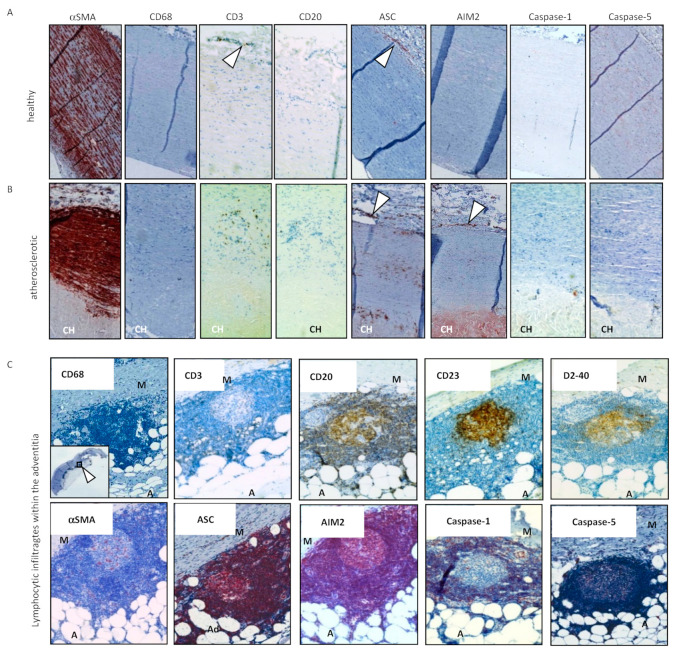
Immunohistochemical stainings of transversial sections derived from healthy aortic wall (**A**), atherosclerotic aorta (**B**) and abdominal aortic aneurysm (**C**). Samples were immunohistochemically stained with antibodies directed against the indicated antigens (SMA (smooth muscle actin), CD68 (macrophages), CD3 (T cells), CD20 (B cells), ASC, AIM2, Caspase-1, Caspase-5. Arrows point to the vasa vasorum; CH: cholesterol plaque; A: Adventitia, M: Media. Original magnification A and B: 40×; original magnification C: 200×.

**Figure 4 cells-10-02433-f004:**
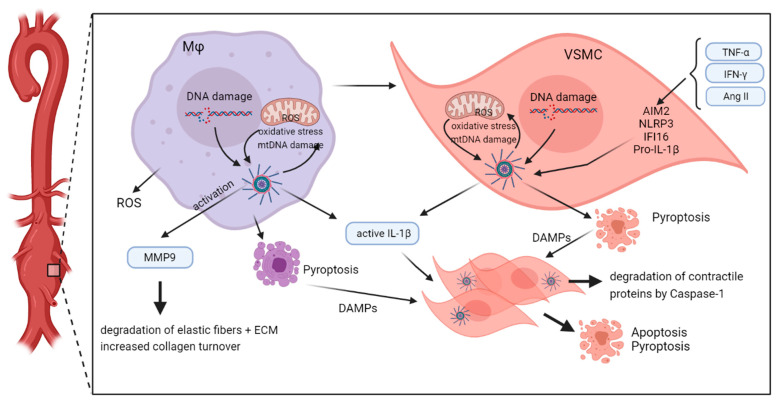
Inflammasome activity within the wall of abdominal aortic aneurysm. Inducers, such as necrotic cell debris, cholesterol crystals, endogenous nuclear, and mtDNA, as well as TNF-α, IFN-γ, or AngII, activate assembly of different inflammasome complexes in resident macrophages and VSMC. In macrophages, this results in activation and release of MMP9, which participates in degradation of elastic fibers and increased collagen turnover. In addition, macrophages release IL-1β and ROS, resulting in response of adjacent VSMC, which on their part, activate inflammasome signaling. Both macrophages and VSMC may undergo pyroptosis or necrotic cell death thereby releasing further cell ingredients that act as DAMPs on neighbouring cells. In the absence of inhibitory factors, this results in a vicious circle of necrotic cell death and release of inflammatory components into the aortic wall. VSMC cell death, remodeling and loss of fibers weaken the aortic wall which may finally end in rupture. The figure was created with BioRender.com (accessed on 15 September 2021).

**Table 1 cells-10-02433-t001:** Expression and function of inflammasome components in cells associated with aortic disease.

Location (Cell Type)	Aortic Disease/Animal Model	Expression/Function	Reference
NLRP3			
Abdominal aorta, (Infiltrating lymphoid cells)	AAA (h)	Strong expression in early AAA stages	[30]
Abdominal aorta (Infiltrating macrophages)	AAA (h)	Strong expression in early AAA stages	[30]
Aortic wall	AAA (h)	Increased gene expression	[30,32]
Thoracic aortic wall, macrophages and VSMC	TAA (h),	Increased protein levels	[44]
VSMC	AD (h)	Increased expression versus healthy, activation of MMP9	[56,62]
Medial VSMC, adventital macrophages	AD (h)	High expression in AD tissue	[62]
PBMC derived macrophages	Takayasu arteritis (h)	Increased versus healthy controls	[89]
PBMC and skin lesions	Behçet’s syndrome (h)	Increased versus healthy controls	[91]
Primary VSCM	In vitro	Induced by β-Glycerophosphate	[46]
AAA-derived VSMC	In vitro	Induced by necrotic cell debris	[47]
mouse VSMC	AngII-induced TAAD mouse model	Gene deletion attenuates VSMC transformation	[48]
Adventitial macrophages	AngII-induced TAAD mouse model in Apoe^−/−^	Genetic deficiency decreases incidence, maximal diameter and severity of aortic aneurysm Mitochondrial oxidative stress	[50]
Adventitial macrophages	Hyperhomocystemia-aggravated AAA in AngII-induced mouse model and in calcium phosphate-induced mouse model	Mitochondrial oxidative stress	[51]
VSMC	AngII-induced mouse model (Nnt deficient)	Increased versus NNT-proficient cells/animals; mitochondrial oxidative stress	[53]
VSMC	TAAD; AngII-induced mouse model + high fat diet	Increased degradation of myosin heavy chain and tropomyosin	[44,55]
Femoral artery	Thrombotic occlusion in a rat model for Buerger’s disease (h)	Increased expression	[87,88]
Aortic wall	Mouse model for Kawasaki disease	deficiency inhibited AAA formation,	[94]
**NLRP1**			
Aortic full vessel wall	AOD (h)	Increased versus healthy controls and AAA	[84]
**AIM2**			
Infiltrating lymphoid cells	AAA (h)	Strong expression in early AAA stages	[29,30]
Infiltrating macrophages	AAA (h)	Strong expression in early AAA stages	[29,30]
EC	AAA (h)	Inducible by different stimuli	[29]
VSMC	AAA (h)	Inducible by different stimuli	[29]
PBMC of vascular patients	AAA (h)	Increased in AAA versus non-AAA	[35]
Peripheral granulocytes, monocytes, lymphocytes	AAA (h)	Increased in AAA versus non-AAA	[34]
Aortic wall, media	AD (h)	Increased expression versus healthy	[63]
Primary EC, VSMC	In vitro	Induced by TNF-α, IFN-β, poly-dA:dT,	[29]
AAA-derived VSMC	In vitro	Induced by necrotic cell debris	[47]
VSMC (m)	AngII-induced mouse model, in vitro	Genetic deficiency results in reduced viability and stronger calcification of VSMC, reduced incidence of AA	[53]
VSMC	AngII-induced mouse model (Nnt deficient)	Increased versus NNT-proficient cells/animals; mitochondrial oxidative stress	[54]
**IFI16**			
Aortic wall	AAA (h), AOD (h)	Increased gene expression	[31,32]
AAA-derived VSMC	In vitro	Induced by necrotic cell debris	[47]
AAA-derived VSMC	In vitro	Induced by AngII	[31]
**ASC**			
Infiltrating lymphoid cells	AAA (h)	Strong protein expression	[29,30]
Thoracic aortic wall, macrophages and VSMC	TAA (h)	Strong protein expression	[44]
PBMC and skin lesions	Behçet’s syndrome	Increased versus healthy controls	[91]
Primary VSCM	In vitro	Induced by β-Glycerophosphate	[46]
AAA-derived VSMC	In vitro	Induced by Angiotensin II	[31]
Adventitial macrophages	AngII-induced TAAD model in Apoe^−/−^ mice	Genetic deficiency decreases incidence, maximal diameter and severity of aortic aneurysm	[50]
VSMC	AngII-induced mouse model (Nnt deficient)	Increased versus NNT-proficient cells/animals; mitochondrial oxidative stress	[53]
**Pro-Caspase-1**			
PBMC derived macrophages	Takayasu arteritis	Increased versus healthy controls	[89]
PBMC and skin lesions	Behçet’s syndrome	Increased versus healthy controls	[91]
Primary VSCM	In vitro	Induced by β-Glycerophosphate	[46]
AAA-derived VSMC	In vitro	necrotic cell debris	[47]
VSMC	AngII-induced mouse model (Nnt deficient)	Increased versus NNT-proficient cells/animals; mitochondrial oxidative stress	[53]
VSMC	AngII-induced mouse model + high fat diet	Increased degradation of myosin heavy chain and tropomyosin	[44]
Aortic wall	Mouse model for Kawasaki disease	deficiency inhibited AAA formation,	[94]
**Active Caspase-1**			
Infiltrating lymphoid cells	AAA (h)	Strong protein expression	[29,30]
PBMC of vascular patients	AAA (h)	Increased in AAA versus non-AAA	[35]
Thoracic aortic wall, macrophages and VSMC	TAA (h)	Strong protein expression	[44]
Adventitial macrophages	AngII-induced TAAD mouse model in Apoe^−/−^	Genetic deficiency decreases incidence, maximal diameter and severity of aortic aneurysm	[50]
Bone marrow derived dendritic cells	Mouse model for Kawasaki disease	Caspase-1 activation, which was inhibited by deficiency in IL-1β, NLRP3, ASC or by antibody targeting IL-1β	[95]
**Caspase-5**			
Infiltrating lymphoid cells	AAA (h)	Strong protein expression	[29]
**IL-1β**			
PBMC of vascular patients	AAA (h)	Increased in AAA versus non-AAA	[34]
Plasma of vascular patients	AAA (h)	Increased in AAA versus non-AAA	[34]
Thoracic aortic wall	TAA (h), TAAD (m)	Increased levels versus healthy aorta	[42,43]
Aortic wall, Media	AD (h)	Increased versus aortic wall from healthy individuals and versus AOD patients	
Supernatant of PBMC derived macrophages	Takayasu arteritis	Increased release versus healthy controls	[89]
PBMC	Behçet’s syndrome, in vitro	Increased release versus healthy controls	[91]
EC and VSMC	in vitro	Induced by different stimuli	[45]
Supernatant of primary VSCM	In vitro	Induced by β-Glycerophosphate	[46]
systemic	Elastase induced TAA mouse model	Deletion and pharmacological inhibition reduce TAA formation	[43]
systemic	AngII-induced TAAD mouse model	Pharmacological inhibition reduces TAA formation	[49]
systemic	BAPN-induced mouse model for TAAD	Activation of PKM2 suppresses NLRP3 mediated IL-1β secretion	[57]
**IL-18**			
Plasma	AD (h)	Increased levels	[64]
aortic samples	AD (h)	Increased levels	[64]
Macrophages,	AD (h)	Activation and release results in altered macrophage polarization and aortic infiltration	[64]
CD4+ lymphocytes, VSMC	AD (h)	Regulation of macrophage induced SMC apoptosis	[64]
systemic	AngII infused mouse model + BAPN	Genetic deficiency decreases incidence, maximal diameter and severity of aortic aneurysm	[52]

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
