# Peer review of "Inflammasomes in the Pathophysiology of Aortic Disease"

_cells, 2021, doi:10.3390/cells10092433_

Round 1

Reviewer 1 Report

This is an excellent review article regarding the description of types of inflammasomes implicated in different types of aortic disease. The authors have put serious effort to summarize the published reports.

Overall, the manuscript addresses an interesting topic and the contents are well-organized. However, its structure and presentation need minor revision. The authors should consider the abbreviations written as NLRP-1/NLRP-3. The authors should be consistent with their abbreviation Ang-II/Ang II in the text and table and should not mix between them within the manuscript.

Comments:

  1. As indicated in line no. 67-68, the activation of procaspase-1 leads to the activation of pro-interleukins (IL) -1β and -18. However, activation of procaspase -1 also activates IL-33. The authors need to include the IL-33 as well.

  1. As indicated in line no 71, several caspases, such as caspase- 4, -5, and -11, are involved in non-canonical inflammasomes activation. In addition, with these caspases, Caspase-8 is involved in non-canonical inflammasomes activation. Authors need to add the involvement of caspase-8 activation in non-canonical inflammasomes activation.

  1. Table 1 presents the studies of inflammasomes components carried out in vitro, isolated cells, and animals. The table should be divided into two parts including clinical/in vivo data and in vitro/ isolated cells.

  1. Table 1 summarizes the inflammasomes components expressed/induced in disease conditions/inducers. The folds of induction of inflammasomes components are also important and the information should be provided if possible. Moreover, the functional role of inflammasomes is also critical, which is also missing in Table 1. The authors should include the functional role of inflammasomes in Table 1.

  1. The authors describe the role of IL-1β in the pathophysiology of aortic disease, however, the authors did not summarize the role of IL-18 in aortic disease. Few reports have described the role of IL-18 in aortic disease. The authors need to summarize and include such reports in the manuscript and table.

  1. The sentence of line no. 150 is incomplete.

  1. Some words wrote in figure. 4 are very small to understand them. The authors should enlarge the font so that it would be easy to read and understand.

  1. Reports have shown that inflammasomes activation has shown to induce both apoptotic or pyroptotic cell death. The authors should specify the type of cell death in particular experimental conditions if possible.

Author Response

Point by point response to reviewer 1:

Comments and Suggestions for Authors

This is an excellent review article regarding the description of types of inflammasomes implicated in different types of aortic disease. The authors have put serious effort to summarize the published reports.

Overall, the manuscript addresses an interesting topic and the contents are well-organized. However, its structure and presentation need minor revision. The authors should consider the abbreviations written as NLRP-1/NLRP-3. The authors should be consistent with their abbreviation Ang-II/Ang II in the text and table and should not mix between them within the manuscript.

Response: We thank the reviewer for these comments. All abreviations have been ckecked for consistency within the text and the table.  Upper case or lower case letters and italic writing has been performed according to international standards (Upper case for human genes and protein names, first letter upper case for mouse genes and proteins, italics for genes and non-italic for proteins).

  1. As indicated in line no. 67-68, the activation of procaspase-1 leads to the activation of pro-interleukins (IL) -1β and -18. However, activation of procaspase -1 also activates IL-33. The authors need to include the IL-33 as well.

 Response 1:  We thank the reviewer for this comment. Because reviewer 2 had a similar comment, the sentence in line 68/69 has been changed accordingly, “…and several other cytokines of the IL-1 family.”

2. As indicated in line no 71, several caspases, such as caspase- 4, -5, and -11, are involved in non-canonical inflammasomes activation. In addition, with these caspases, Caspase-8 is involved in non-canonical inflammasomes activation. Authors need to add the involvement of caspase-8 activation in non-canonical inflammasomes activation.

 Response 2: Again, reviewer 2 had a similar comment. As suggested, the sentence was changed and caspase-8 was included together with an additional reference (17).

3. Table 1 presents the studies of inflammasomes components carried out in vitro, isolated cells, and animals. The table should be divided into two parts including clinical/in vivo data and in vitro/ isolated cells.

 Response 3: We appreciate the reviewer’s comment. However, we would like to leave the table undivided. In contrast to the text, which is arranged in chapters according to different aortic diseases, the table is arranged according to single inflammasome proteins, to give an overview about clinical, in vivo and in vitro data, referring to each component. For better understanding, the table was reformatted (visible in the version without tracking)

4. Table 1 summarizes the inflammasomes components expressed/induced in disease conditions/inducers. The folds of induction of inflammasomes components are also important and the information should be provided if possible. Moreover, the functional role of inflammasomes is also critical, which is also missing in Table 1. The authors should include the functional role of inflammasomes in Table 1.

Response 4: We thank the reviewer for this comment. The fold change of induction is only available from a few publications, and differs greatly between cells tissues and experiments within these publications. In addition, the fold change of expression might be related to the experimental conditions and is not very informative considering the function in a dynamic process that changes within a few minutes. The terms “increased“ or “reduced” show that the process has been started or attenuated, which is the main information. Considering the functional role, we have included the function of the different components whenever possible in the table. However, the function of the individual components is not necessarily equal to the function of the complete inflammasome in a certain situation. Therefore, these details have been explained in the text as precise as possible.

5. The authors describe the role of IL-1β in the pathophysiology of aortic disease, however, the authors did not summarize the role of IL-18 in aortic disease. Few reports have described the role of IL-18 in aortic disease. The authors need to summarize and include such reports in the manuscript and table.

 Response 5: We have included the role of IL-18 in line 274 (mouse model for a role in AAA) and 370 (clinical data for a role in AD), together with two additional references.

 6. The sentence of line no. 150 is incomplete.

Response 6: We apologize for missing parts in the manuscript. Several sentences have been rewritten.

7. Some words wrote in figure. 4 are very small to understand them. The authors should enlarge the font so that it would be easy to read and understand.

 Response 7: Figure 4 has been substituted by a new figure with larger fonts.

8. Reports have shown that inflammasomes activation has shown to induce both apoptotic or pyroptotic cell death. The authors should specify the type of cell death in particular experimental conditions if possible.

 Response 8: Whenever possible we have specified these differences in the text, e.g. line. It should be noticed that inflammasome activity does not necessarily lead to cell death, neither apoptotic nor pyroptotic. According to our summary, the major function of inflammasome activation within a cell is to inform neighbouring cells ant inflammatory cells by release of cytokines (IL-1) about an internal damage that needs to be recognized. This release is also possible without cell death as described in chapter 2.

Reviewer 2 Report

Comments on Wortmann et al “Inflammasomes in the pathophysiology of aortic disease”.

The review is well written and covering an important area of interest for both clinicians and basic scientists.

My recommendation is “Accept for publication following minor revision”.

Comments:

  • The concept of a causal role of inflammasomes in aged-related aortic diseases would be misleading to a half of the truth. This review would be more comprehensive if the authors could provide an in-depth discussion on functional interplay between molecular changes in aging, and the inflammasome signaling on the other end. The authors have had some discussion on cholesterol crystal-driven NLRP3 inflammasome activation as a causative factor in aortic atherosclerosis and occlusive aortic disease. Other senescent pathways could also be involved in aortic pathology e.g. elastin breakdown, accumulation of advanced glycation end products/their receptors (AGE/RAGE), lysosome accumulation of lipid peroxidation adducts, the senescence-associated secretory phenotype etc. Importantly, interactions could be mutually causative in self-perpetuating processes, not just inflammasome activation causative to aged-related diseases. Reports from non-vascular studies indicate that elastase effects could be causative to inflammasome activation (Couillin et al J Immunol 2009), elastin fragments could induce NFkB activation and IL-1beta upregulation – a prime condition for inflammasome activation (Debret et al J Invest Dermatol 2006) etc. Increased AGE/RAGE signaling, another senescence-associated feature frequently found in vasculature including aorta, was reported linked to the NLRP3 inflammasome activation (Jia et al Cell Death Dis 2019).      
  • Line 57: “inflammasomes form high-molecular weight signalling platforms” should be corrected to “inflammasome proteins aggregate to form (cytosolic) high-molecular weight signalling platforms which are called inflammasomes”.
  • Line 68: To be precise, inflammasomes cleave/activate not only IL-1beta and IL-18 but a few other less studied cytokines of the IL-1 family. Please correct.
  • Lines 70-72: “In addition to the canonical inflammasome signalling, other inflammatory caspases such as caspase-4, -5, and -11…” should be corrected to “In addition to the canonical inflammasome signalling, non-canonical inflammasomes involving other inflammatory caspases such as caspase-4, 5-, and -11…”
  • Line 229: “in other vascular cells”, remove OTHER.
  • Line 435 “occlusive aortic disease”, change to “aortic occlusive disease” to be consistent with the abbreviation AOD.
  • Last paragraph is utmost important. Its first sentence reads very bumpy, can you please rewrite.
  • Very minor/style: the comma could be omitted in a number of sentences e.g. line 81 (“introduced, here.”), line 88 (“guarantee both,”), line 290 (“in both, macrophages and VSMC”), line 553 (“For both, acute and chronic aortic disease”, line 573 (“The picture, how”, line 578 (“further details, that”)…

Author Response

We thank the reviewer for useful comments and suggestions. Please find the point by point response below.

Reviewer 2

  • The concept of a causal role of inflammasomes in aged-related aortic diseases would be misleading to a half of the truth. This review would be more comprehensive if the authors could provide an in-depth discussion on functional interplay between molecular changes in aging, and the inflammasome signaling on the other end. The authors have had some discussion on cholesterol crystal-driven NLRP3 inflammasome activation as a causative factor in aortic atherosclerosis and occlusive aortic disease. Other senescent pathways could also be involved in aortic pathology e.g. elastin breakdown, accumulation of advanced glycation end products/their receptors (AGE/RAGE), lysosome accumulation of lipid peroxidation adducts, the senescence-associated secretory phenotype etc. Importantly, interactions could be mutually causative in self-perpetuating processes, not just inflammasome activation causative to aged-related diseases. Reports from non-vascular studies indicate that elastase effects could be causative to inflammasome activation (Couillin et al J Immunol 2009), elastin fragments could induce NFkB activation and IL-1beta upregulation – a prime condition for inflammasome activation (Debret et al J Invest Dermatol 2006) etc. Increased AGE/RAGE signaling, another senescence-associated feature frequently found in vasculature including aorta, was reported linked to the NLRP3 inflammasome activation (Jia et al Cell Death Dis 2019).   

Response 1:  We thank the reviewer for these detailed comments. As suggested, we have included an additional paragraph in the final chapter 9, to discuss these issues. Moreover, additional references were included, where necessary.

  Line 57: “inflammasomes form high-molecular weight signalling platforms” should be corrected to “inflammasome proteins aggregate to form (cytosolic) high-molecular weight signalling platforms which are called inflammasomes”.

Response 2: The sentence was corrected as suggested.

  • Line 68: To be precise, inflammasomes cleave/activate not only IL-1beta and IL-18 but a few other less studied cytokines of the IL-1 family. Please correct.

Response 3: A similar comment was made by reviewer 1. We have changed the sentence accordingly.

  • Lines 70-72: “In addition to the canonical inflammasome signalling, other inflammatory caspases such as caspase-4, -5, and -11…” should be corrected to “In addition to the canonical inflammasome signalling, non-canonical inflammasomes involving other inflammatory caspases such as caspase-4, 5-, and -11…”

Response 4: A comment to this sentence was also made by reviewer 1. We have changed the sentence accordingly.

  • Line 229: “in other vascular cells”, remove OTHER.

Response 5: The sentence was corrected accordingly.

  • Line 435 “occlusive aortic disease”, change to “aortic occlusive disease” to be consistent with the abbreviation AOD.

Response 6: The term was corrected accordingly in both the text and in new figure 2.

  • Last paragraph is utmost important. Its first sentence reads very bumpy, can you please rewrite.

Response 7: The sentence was rewritten, as suggested

  • Very minor/style: the comma could be omitted in a number of sentences e.g. line 81 (“introduced, here.”), line 88 (“guarantee both,”), line 290 (“in both, macrophages and VSMC”), line 553 (“For both, acute and chronic aortic disease”, line 573 (“The picture, how”, line 578 (“further details, that”)…

 Response 8: Commas in the text were corrected .